# Data-driven discovery and model reduction methods for the atmospheric effects of high altitude emissions

Jurriaan A. van 't Hoff<sup>1</sup>, Tom S. van Cranenburgh<sup>1</sup>, Urban Fasel<sup>2</sup>, and Irene C. Dedoussi<sup>1,3</sup>

Correspondence: Irene C. Dedoussi (icd23@cam.ac.uk)

#### Abstract.

Chemistry transport models play a crucial role in the evaluation of the effect of anthropogenic emissions on the atmosphere and climate, but they come with high computational costs and require specialized know-how. This renders them impractical for applications in multidisciplinary optimisation, or regulatory and operational-decision making processes where environmental effects are to be considered. Such applications require computationally efficient surrogate models of the complex chemistry transport models. Here we investigate the use of data-driven discovery and reduced-order modelling methods for this purpose. Specifically, we examine the dynamic mode decomposition (DMD) and proper orthogonal decomposition coupled with the sparse identification of non-linear dynamics (POD-SINDy). We evaluate their ability to reconstruct and forecast changes in the distribution of ozone in response to the introduction of supersonic aircraft as modelled by the GEOS-Chem chemistry transport model. Of the tested methods, we find that optimized DMD and bagging optimized DMD perform best. These methods can reconstruct and forecast full-atmospheric ozone responses for up to several years without losing stability, at smaller errors than estimates using the spatio-temporal mean of the data. On average, the optimized DMD method reduces the reconstruction error by 55.2% and that of forecasting by 19.4%. For the bagging optimized DMD these reductions are 40.3% and 7.9%, respectively. The resulting change in global ozone column, calculated from the reconstructed atmospheres, has an error smaller than 10%. This is achieved while reducing the computational and storage requirements by several orders of magnitude, which may be a worthwhile tradeoff for some applications.

#### 1 Introduction

Chemistry transport models (CTMs) and climate chemistry models (CCMs) are critical to our understanding of how anthropogenic emissions affect the environment and climate. These models simulate the chemistry, transport, and deposition of hundreds of chemical species in the atmosphere under evolving meteorological conditions, allowing them to capture complex chemical feedback mechanisms and responses. This level of complexity requires specialised expertise and comes with a considerable computational cost, which is why these models usually require access to dedicated high-performance computer systems. Despite this barrier, CTMs and CCMs are widely used to study a variety of problems in atmospheric composition and

<sup>&</sup>lt;sup>1</sup>Operations & Environment, Faculty of Aerospace Engineering, Delft University of Technology, Kluyverweg 1, 2629 HS, Delft, The Netherlands

<sup>&</sup>lt;sup>2</sup>Department of Aeronautics, Faculty of Engineering, Imperial College London, Exhibition Road, SW7 2AZ, London, UK <sup>3</sup>Department of Engineering, University of Cambridge, 1 JJ Thomson Avenue, Cambridge, CB3 0DY, Cambridge, UK

chemistry, although their use is often limited to the evaluation of a handful of case-studies.


In the consideration of new technologies, such as new aircraft concepts, multiple parties have interest in the evaluation of CTMs and CCMs. Engineers may want to integrate their environmental assessment into multidisciplinary optimisation (MDO) approaches to minimise environmental effects in their design process. Regulatory bodies may be interested in evaluating the effectiveness of certain regulations, and operators may want to know if environmental effects can be reduced through operations. These parties often take iterative approaches, that may require up to hundreds or thousands of cycles over their applications. Considering that CTMs and CCMs often require days, if not weeks, to perform their evaluations, they are impractical to integrate in these approaches, even if the technical know-how to use them is already present. Therefore, to support the integration of environmental evaluations into MDO approaches or regulatory and operational decision-making processes, we need easy-to-use alternatives to replicate or estimate the output of these models at a fraction of their computational cost.


The recent developments in the field of machine learning have led to new methods that may be suitable for low-cost surrogate models, but many of them (e.g. neural networks) require vast amounts of data, which is impractical to generate with CTMs and CCMs. Data-driven model discovery and reduced-order modelling methods, that require less data to operate and are more interpretable than "black-box" neural network models, can be suitable alternatives. Two methods of interest are Dynamic Mode Decomposition (DMD) and Proper Orthogonal Decomposition (POD) coupled with the Sparse Identification of Non-Linear Dynamics (SINDy). These methods extract low-dimensional features from data that can be used for analysis and to construct reduced-order models, for which they are already widely used in the field of fluid mechanics (e.g. Taira et al., 2017, 2020; Champion et al., 2019; Khodkar and Hassanzadeh, 2021; Callaham et al., 2022). Atmospheric processes share some similarities with fluid flows, and earlier work has shown that these methods may be transferable. For example, Yang et al. (2024) have shown that dimensionality reduction techniques can be combined with SINDy to reproduce atmospheric data from CTMs, and Velegar et al. (2024) have shown that DMD methods can be used to produce efficient reduced-order models to forecast the concentration of several tropospheric chemicals over the course of multiple months. These initial explorations show promising results, but the application of the methods is still limited to small subsets of CTM data.

p

We expand on this exploration by assessing the suitability of the DMD and POD-SINDy methods on large-scale and complex datasets that describe chemistry and transport processes across the entire atmosphere. Where earlier work assessed the capability of these methods to reproduce parts of unperturbed atmospheres, we evaluate their use on data describing perturbations over the entire atmosphere. Specifically, we apply the methods to datasets that describe the difference of the ozone (O<sub>3</sub>) distribution in response to different supersonic aircraft scenarios (van 't Hoff et al., 2025b, 2024b). The aircraft attributable ozone effect is identified as the difference between two independent CTM evaluations (one with supersonic aircraft emissions, and one without). If data-driven methods can reconstruct and predict these differences, they may be directly usable for the estimation of environmental effects in MDO applications or scenario analyses. We assess the performance of these methods through a stepwise approach, starting from their interpretability for the analysis of atmospheric processes, followed by their

reproductions of data in specific locations and the entire atmosphere. We also assess the possibility of calculating other metrics of from the reproduced atmosphere, a capability that may be valuable for several applications.

## 2 Case studies

We use seven datasets that describe the changes in global ozone distribution in response to the adoption of supersonic aircraft. These datasets were generated using the Dutch national supercomputer Snellius using an estimate of around 493,920 CPU-hours. From van 't Hoff et al. (2024b) we use four datasets which describe the response to 8 Tg of annual emissions from different flight corridors. These are the transatlantic flight corridor (TAC) and the corridor over the southern Arabian sea (SAS). In both cases we use datasets with emission altitudes of 16.2 and 20.4 km. These are therefore denoted as SAS204, TAC204, SAS162, and TAC162 respectively. These datasets are generated with version 13.3.1 of the GEOS-Chem CTM and represent 10-year of daily changes in global ozone mass at a horizontal resolution of  $4^{\circ} \times 5^{\circ}$  (latitude  $\times$  longitude) with 72 vertical pressure levels.

We also use three other datasets from van 't Hoff et al. (2025b). These also describe changes in the distribution of the ozone mass over 10 years, but in response to emissions from a global supersonic aircraft fleet rather than from a regional source. Due to the global distribution of these emissions we expect that these datasets may exhibit more challenging dynamical behaviour compared to the regional emission datasets. These datasets are generated with version 14.3.0 of the high-performance GEOS-Chem model, with a horizontal resolution of  $2^{\circ} \times 2.5^{\circ}$  and 72 vertical pressure levels. Following the notation of van 't Hoff et al. (2025b) we denote these datasets as S1, S2, and S3. These represent the change in ozone in response to a theoretical supersonic fleet operating at Mach 2.0 with cruise altitudes of 16.5 to 19.5 km, one with increased  $NO_x$  emissions, and one with a lower cruise altitude and speed (Mach 1.6, 14-16.5 km), respectively. Detailed descriptions of the emission scenarios and chemistry transport model setups are provided in the related works (van 't Hoff et al., 2024b, 2025b).

# 80 3 Methodology


The methodology for this work is outlined in Figure 1. Subsequent subsections will discuss the preprocessing of data and the model discovery methods.

# 3.1 Data description and processing

The datasets that we use contain four-dimensional (time, longitude, altitude, latitude) descriptions of the daily-averaged changes in ozone mass over the course of ten years. We take several pre-processing steps to reduce the complexity of our data. All datasets exhibit an initial transient response to the introduction of the supersonic aircraft emissions, followed by a stabilized response where the new atmospheric quasi-equilibrium is reached. We first isolate the stable response from the data by removing the transient part, which spans the first five years. Secondly, we reduce the dimensionality of the dataset

**Figure 1.** Methodology overview. In step 1 the effects of emissions are isolated by combining two GEOS-Chem atmosphere simulations, of which one is affected by the additional emissions. The resulting spatiotemporal datasets describe the daily average changes in ozone mass distribution for a period of 10 years. In step 2 the data is preprocessed and organised as a snapshot of matrices prior to applying data-driven methods (step 3). In step 4 the data-driven methods are used to forecast future behaviour of the ozone response, based on which the data-driven methods are assessed.

by calculating the longitudinally-averaged ozone response. Longitudinal averaging is standard practice in studies of the environmental effects of supersonic aircraft, because in the multi-annual timespans considered, the atmosphere may be considered as well-mixed over the longitude (Zhang et al., 2021a, 2023; Eastham et al., 2022). Thirdly, the GEOS-Chem model uses a non-uniform vertical grid that is denser near the surface. This is not uncommon for CTMs, but in this case study the majority of the ozone response occurs in the coarser stratospheric grid. To avoid over-representation of the near-surface conditions in the data, we therefore interpolate the data to a normalized vertical grid with a resolution of 1 km. Finally, we discard the grid cells above 51 km altitude, as this is the vertical limit of GEOS-Chem's extensive chemical solver for ozone (Eastham et al., 2014).

The data may be considered as combination of a dominant steady-state change in the ozone distribution and a dynamic component that is driven by seasonal and day-to-day changes in meteorology. Across all datasets, the steady-state response exhibits similar characteristics, with increases of ozone in the lower stratosphere and depletion of ozone in the upper stratosphere, yielding a net-loss of the global ozone column. For detailed explanations of the mechanisms behind these responses, we refer to the related articles (van 't Hoff et al., 2024b, 2025b). Figure 2 shows the average steady state for one of the datasets (TAC204) alongside four seasonal averages, highlighting that the ozone response shifts seasonally.

**Figure 2.** Average ozone response of the TAC204 dataset over different timespans. The leftmost plot shows the average response over all 5 years of data, and the other four show averages taken over 3-monthly snapshots from the entire dataset. For example, the rightmost figure (Average (O N D)) shows the average ozone response for the months of October, November, and December across the 5 years of data.

This shift is part of the dynamic component, that we isolate by subtracting the temporal mean of the data similar to Velegar et al. (2024) and Yang et al. (2024). This mean is re-added after the dynamics are predicted to recompose the full signal. The data is organised in a matrix of snapshots **X**:

$$\mathbf{X} = \begin{bmatrix} | & | & | \\ \mathbf{x}(t_1) & \mathbf{x}(t_2) & \cdots & \mathbf{x}(t_m) \\ | & | & | \end{bmatrix}$$
 (1)

The matrix  $\mathbf{X} \in \mathbb{R}^{n \times m}$  represents the ozone response, with n representing the product of the latitude and altitude dimensions and m the number of daily snapshots. In this representation, the latitude and altitude are flattened into a single column  $x(t_y)$ . To evaluate the modelling of the entire atmosphere, we also calculate the mean global change in the ozone columns in Dobson Units (DU). This metric is also commonly used in studies of the environmental effects of supersonic aircraft (Zhang et al., 2023, 2021a, b; Eastham et al., 2022; van 't Hoff et al., 2025b).

## 3.2 Proper orthogonal decomposition

The POD originates from applications in the field of fluid mechanics and it uses the singular value decomposition (SVD) in the context of spatiotemporal data (Brunton and Kutz, 2021; Weiss, 2019). Applied to spatiotemporal data, the POD returns spatial patterns based on the variance in the data, also referred to as modes, and an associated temporal evolution of these modes. The spatial modes are hierarchically ordered, making truncated POD approximations of the high-dimensional data straightforward by isolating the first r spatial modes. The POD seeks to find an approximation of the mass distribution of ozone X at spatial

dimension x (both latitude and altitude) and time t:

120 
$$\mathbf{X}(x,t) = \overline{\mathbf{X}}(x) + \sum_{k=1}^{\tau} a_k(t)\psi_k(x),$$
 (2)

where  $\overline{\mathbf{X}}(x)$  is the temporal mean of ozone response,  $\psi_k(x)$  the  $k^{th}$  POD spatial mode, and  $a_k(t)$  the POD temporal coefficient of the  $k^{th}$  mode at time t. By combining the product of the spatial and temporal coefficients up to a certain rank r, reduced-order representations of the original data can be reconstructed.

## 3.3 Sparse identification of non-linear dynamics (SINDy)

To forecast future behaviour, the future state of the temporal coefficient needs to be forecast, and for this we use sparse identification of non-linear dynamics (SINDy) to find a dynamical system model for the temporal coefficients. The SINDy method can be applied to data to discover ordinary differential equations (ODEs) and partial differential equations (PDEs) which describe its dynamics (Brunton et al., 2016; Rudy et al., 2017). SINDy has been applied to a range of model discovery problems, including problems in fluid dynamics (Loiseau and Brunton, 2018), computational chemistry (Boninsegna et al., 2018a; Yang et al., 2024), and climate models (Zanna and Bolton, 2020). It is often assumed that the equations that govern these systems are sparse and have only a few active terms in the dynamics, therefore SINDy uses sparse regression to identify the active terms out of a library of candidate model terms. We use SINDy to find analytical models for the dynamics of the temporal coefficients of the POD ( $\mathbf{a}_k(t)$  in Equation 2). SINDy attempts to identify a system of first-order ODEs of the form:

$$\frac{\mathrm{d}}{\mathrm{d}t}\mathbf{a} = \dot{\mathbf{a}} = \mathbf{f}(\mathbf{a}) \approx \mathbf{\Theta}(\mathbf{a})\mathbf{\Xi},\tag{3}$$

where the temporal coefficients of the POD ( $\mathbf{a} = [a_1, a_2, \cdots, a_r]^T$ ) are the states of the dynamical system, and the derivatives ( $\dot{\mathbf{a}}$ ) are approximated as a linear combination of nonlinear functions ( $\mathbf{\Theta}(\mathbf{a})$ ) in a library with coefficients ( $\mathbf{\Xi}$ ). The dynamics ( $\mathbf{f}$ ) of the coefficients can be represented with a library of components. This library can be made up of any set of functions, with low-order polynomial terms commonly being used as a starting point. To identify an accurate and sparse model, the number of nonzero terms in  $\mathbf{\Xi}$  should be as small as possible. To find  $\mathbf{\Xi}$ , a penalized least-squares problem is solved using the sequentially thresholded least-squares algorithm (STLSQ)(Brunton et al., 2016):

$$\mathbf{\Xi} = \min_{\hat{\mathbf{\Xi}}} \left\| \dot{\mathbf{P}} - \mathbf{\Theta}(\mathbf{P}) \hat{\mathbf{\Xi}} \right\|_{2}^{2} + \lambda \|\hat{\mathbf{\Xi}}\|_{0}, \tag{4}$$

Here  $\mathbf{P} = [\mathbf{a}(t_1), \mathbf{a}(t_2), \cdots, \mathbf{a}(t_m)]$  is the POD coefficient matrix, and  $\dot{\mathbf{P}}$  is its derivative. STLSQ sequentially solves a least squares problem and includes a sparsity-promoting threshold  $\lambda$ , setting terms below the threshold to zero, approximating the  $L_0$  norm in Equation 4. To find suitable SINDy models, we therefore perform a sequential grid search over a range of threshold values, using the corrected Aikaike information criteria (AIC-c) (Mangan et al., 2017) to optimize the model selection for sparsity.



The SINDy method uses the derivative of the data, making the method susceptible to noise. We therefore take additional pre-processing steps to improve the robustness of the SINDy method when applied to the temporal coefficients. We first reduce noise in the coefficients using a Savitzky-Golay filter (Savitzky and Golay, 1964) with a 60-day temporal window. Following this, the values of the time coefficients are normalized, and finally the time coefficients are temporally interpolated by a factor ten. These steps reduce the magnitude and variability of the derivatives, making them easier to model using the SINDy method. In addition to this pre-processing, we also apply the weak and ensembling forms of the SINDy algorithm that are more suitable for dealing with noisy data (Reinbold et al., 2020; Messenger and Bortz, 2021; Fasel et al., 2022). We use a sequential grid search over a range of threshold values to identify optimal sparse formulations based on the AIC-c. We do this for the standard SINDy, the ensemble, and the weak formulations. These methods are incorporated in the PySINDy Python package (de Silva et al., 2017; Kaptanoglu et al., 2021b).

## 3.4 Dynamic mode decomposition

The dynamic mode decomposition (DMD) decomposes multidimensional spatiotemporal data into spatially coherent structures with linear temporal behaviour, each with an associated frequency of oscillation and rate of growth or decay. For the DMD method, two matrices of snapshots are arranged ( $\mathbf{X}$  and  $\mathbf{X}'$ ). These are similar to the  $\mathbf{X}$  matrix of the POD method, but these matrices omit the final and first snapshots of the data respectively:

$$\mathbf{X} = \begin{bmatrix} | & | & | \\ x(t_1) & x(t_2) & \cdots & x(t_{m-1}) \\ | & | & | \end{bmatrix}$$
 (5)

$$\mathbf{X}' = \begin{bmatrix} & | & & | \\ \mathbf{x}(t_2) & \mathbf{x}(t_3) & \cdots & \mathbf{x}(t_m) \\ & | & & | & & | \end{bmatrix}$$
 (6)

The DMD algorithm finds the best-fit linear operator A that relates these snapshot matrices.

$$\mathbf{X}' \approx \mathbf{A}\mathbf{X} \tag{7}$$

The following steps are followed when applying DMD:

1. The truncated singular value decomposition of matrix  $\mathbf{X}$  is calculated, where  $\tilde{\mathbf{U}} \in \mathbb{C}^{n \times r}$  and  $\tilde{\mathbf{V}} \in \mathbb{C}^{m \times r}$  and  $\tilde{\mathbf{\Sigma}} \in \mathbb{R}^{r \times r}$  depend on the chosen rank r.

$$\mathbf{X} \approx \tilde{\mathbf{U}}\tilde{\mathbf{\Sigma}}\tilde{\mathbf{V}}^* \tag{8}$$

2. Matrix **A** is then obtained by computing the pseudo inverse of **X**. By projecting **A** on the SVD modes **U**, the full matrix **A** does not have to be computed:

$$\tilde{\mathbf{A}} = \tilde{\mathbf{U}}^* \mathbf{A} \tilde{\mathbf{U}} = \tilde{\mathbf{U}}^* \mathbf{X}' \tilde{\mathbf{V}} \tilde{\mathbf{\Sigma}}^{-1}$$
(9)

3. The eigenvalues  $\Lambda$  of the reduced matrix  $\tilde{\mathbf{A}}$  are the same as the eigenvalues of  $\mathbf{A}$ . The eigendecomposition of  $\tilde{\mathbf{A}}$  is performed:

$$\tilde{\mathbf{A}}\mathbf{W} = \mathbf{W}\mathbf{\Lambda} \tag{10}$$

4. The DMD modes  $\Phi$  are reconstructed with the eigenvectors  $\mathbf{W}$  and the time-shifted snapshot matrix of the original data  $\mathbf{X}'$ .

$$\mathbf{\Phi} = \mathbf{X}' \mathbf{V} \tilde{\mathbf{\Sigma}}^{-1} \mathbf{W} \tag{11}$$

To increase the noise robustness of the DMD algorithm, an optimised DMD (OptDMD) was introduced by Askham and Kutz (2018). This was further extended to bagging optimised DMD (BOPDMD) by Sashidhar and Kutz (2022). Both methods are incorporated in the open source Python package PyDMD (Demo et al., 2018; Ichinaga et al., 2024). In this work we assess the applicability of the standard DMD, standard and constrained OptDMD (OptDMD, OptDMD-C), and BOPDMD.

We split the steady-state component of the datasets (the last five years) into two separate sets. We use the first three and a half years to fit the methods and to test their ability to reproduce the original data. The latter one and a half years of data is used to test the methods' ability to forecast the future behaviour of the dynamical system. We evaluate the methods' forecast and reproduction of the ozone response in terms of the root mean squared error (RMSE). For comparison, we consider the RMSE that would be achieved if the temporal mean of the fitting data were used instead for reconstruction and forecasting. We use this for reference as the temporal mean is also commonly used to represent this complex data in publications (van 't Hoff et al., 2024b; Eastham et al., 2022; Zhang et al., 2023; Speth et al., 2021), and it is the simplest method through which future behaviour can be estimated lacking more complex modelling methodologies.

#### 4 Results




We evaluate four different aspects of these reduced-order modelling and model discovery methods. First, we discuss their general application and use for analysis in subsection 4.1. We then evaluate the applicability of these methods in modelling tasks of different dimensionality and complexity. We start by modelling the ozone response in individual grid cells (subsection 4.2), expanding to the modelling of the 2-dimensional zonally average ozone response (subsection 4.3), and finally to the modelling of relevant metrics derived from the global ozone response, specifically the globally averaged ozone column response (subsection 4.4). We note that the results shown in this section do not represent the best models that we found, but they represent the median model performance from around 50,000 variations evaluated in a hyperparameter search (Appendix A, discussed further in section 5).




## 4.1 Analysis of spatial modes

The POD and DMD methods produce spatial modes that represent patterns of dominant spatial features with coupled time coefficient series that describe how the spatial modes evolve over time. For the DMD method, the spatial modes represent patterns with similar temporal frequencies, whereas for the POD method the spatial modes represent features with high spatial correlations. Based on an extensive hyperparameter search (Appendix A), we present results for DMD methods with a rank eight and POD-SINDy methods with rank four. Considering that the DMD modes are coupled conjugate pairs, this results in DMD and POD-SINDy models with similar levels of complexity. Further increases in rank did not yield considerable improvements to the model discovery methods in terms of mean average errors. Examples of the first four spatial and temporal modes from the OptDMD-C and POD methods applied to the TAC204 dataset are shown in Figure 3. Both of these methods identify different spatial modes and time dynamics, with the OptDMD-C method resulting in oscillating time coefficients of different frequencies, whereas POD results in more irregular and complex time coefficients.

**Figure 3.** Spatial modes 1 to 4 and associated time coefficients of the OptDMD-C methods (left) and the POD method (right) extracted from the 3.5 years of fitting data from the TAC204 dataset. Modes of the OptDMD-C method are shown as as pairs due to their nature as conjugate-couples. In both cases, values of the time coefficients are scaled down by a factor of 1 million kg.

The spatial patterns identified by both methods are driven by changes in the ozone response over time. These are a result of the interactions of chemistry and transport processes, that are also affected by meteorological conditions and the distribution of the perturbed emissions in the atmosphere. This overlap of mechanisms makes it challenging to attribute spatial modes, or patterns therein, to any single process. The time coefficients may however inform on some of the drivers behind these modes. For example: the UV radiation that affects the ozone chemistry changes seasonally, driving seasonal fluctuations in the ozone perturbation (also shown in Figure 2). In the northern hemisphere, this causes the average altitude of the positive

lower-stratospheric ozone perturbation to oscillate yearly by up to 2 km (Figure C1). This oscillation is likely captured by the annual modes of the POD and DMD methods, such as modes  $\Phi_{3-4}$  of the OptDMD-C method and modes  $\psi_1$  and  $\psi_2$  of the POD method shown in Figure 3. The spatial modes show that these modes do however capture dynamic behaviour across both hemispheres, suggesting that they may capture multiple phenomena that have similar temporal frequencies. The tendency to capture multiple processes at once limits the usability of POD and DMD modes for analysis on full atmospheric systems, at least for a species as complex as ozone. It may be possible to identify the role of different processes in more restricted case studies, such as the formation of the ozone hole or the surface concentrations of ozone.

#### 4.2 Individual grid cells

The modelling of individual grid cells may be of interest to several case studies, such as forecasting air quality metrics over specific areas. To assess this capability we consider the performance of the model discovery methods in replicating and forecasting the ozone perturbations for several individual grid cells over time. Figure 4 shows the reconstruction and forecast of 3 cells from the TAC204 dataset, located at the same latitude of 22 °N at altitudes of 1.1, 18.3, and 26 km. These grid cells are located in the troposphere, tropopause, and stratosphere respectively, and therefore represent three different dynamical regimes of the atmosphere with different ozone responses.

**Figure 4.** Ozone response in tonnes for 3 grid cells of the TAC204 dataset, with reconstructions and short-term forecasts from the DMD and POD-SINDy methods. Cells shown are located at 22 °N latitude and altitudes of 26.0, 18.3, and 1.4 km from top to bottom respectively. Vertical black lines mark the difference between the reconstruction (solid lines) and forecast (dashed lines).

We find that the methods have different performance in the different atmospheric regimes. The reconstruction and forecast errors are shown in Table B1. In the lower troposphere (1.1 km altitude, 22°N), the ozone response exhibits a stable seasonal

**Table 1.** RMSE (root mean squared error) of the reconstruction (R) and forecast (F) of longitudinally-averaged ozone response in tonnes for the DMD and POD-SINDy methods over the datasets. The Data mean column shows the zonal average reconstruction and forecast RMSE if the perturbation is estimated using the temporal mean from the fitting data. When method's RMSE is lower than the data mean approach, its values are printed in **bold**.

|        |   | DMD   | OptDMD | OptDMD-C | BOPDMD    | POD-SINDv  | POD-SINDy  | POD-SINDy   | Data mean   |
|--------|---|-------|--------|----------|-----------|------------|------------|-------------|-------------|
|        |   | DIVID | Орилип | Оргымы-с | DOI DIVID | 1 OD-SINDy | (ensemble) | (weak form) | Data ilican |
| SAS204 | R | 42.48 | 46.11  | 31.87    | 38.73     | 58.45      | 65.69      | 99.87       | 57.88       |
|        | F | 64.41 | 75.45  | 65.59    | 72.90     | 98.94      | 76.32      | 87.86       | 77.26       |
| TAC204 | R | 27.93 | 32.78  | 18.92    | 28.49     | 40.65      | 40.49      | 39.08       | 47.43       |
|        | F | 51.12 | 55.78  | 49.11    | 62.88     | 55.02      | 55.22      | 47.03       | 59.02       |
| SAS162 | R | 13.05 | 13.93  | 6.69     | 10.21     | 15.97      | 15.88      | 17.16       | 17.51       |
|        | F | 16.49 | 17.75  | 12.93    | 17.44     | 18.53      | 18.62      | 16.36       | 18.03       |
| TAC162 | R | 15.26 | 16.7   | 9.71     | 11.83     | 17.25      | 16.70      | 18.29       | 20.33       |
|        | F | 19.81 | 21.52  | 19.55    | 18.76     | 25.36      | 21.55      | 21.23       | 22.01       |
| S1     | R | 7.07  | 7.20   | 4.55     | 5.82      | 9.26       | 9.22       | 8.60        | 9.79        |
|        | F | 9.10  | 10.78  | 8.96     | 9.67      | 8.98       | 8.87       | 7.66        | 11.45       |
| S2     | R | 23.27 | 23.27  | 12.60    | 17.43     | 37.72      | 37.69      | 897.35      | 28.63       |
|        | F | 32.2  | 32.08  | 23.35    | 27.39     | 32.12      | 30.96      | 506.66      | 32.01       |
| S3     | R | 3.04  | 3.07   | 1.74     | 1.92      | 4.78       | 4.78       | 3.69        | 3.54        |
|        | F | 4.71  | 4.87   | 4.17     | 4.54      | 5.12       | 5.12       | 4.66        | 4.94        |

oscillation, as the response in this domain is primarily driven by the availability of residual UV radiation. This behaviour is generally reconstructed well by most methods, with RMSEs ranging from 0.57 tonnes (BOPDMD) to 1.59 tonnes (POD-SINDy) of ozone. Using the data mean, the reconstruction RMSE is 1.9 tonnes of ozone at this altitude, meaning that in this case all data-driven methods provide an improvement over the propagation of the data-mean.

While the methods offer better reconstructions on average, most of them are not numerically stable. The DMD, OptDMD, and POD-SINDy methods all start to deviate after half a year. Most stagnate after around 18 months of integration, whereas the weak form POD-SINDy overshoots the ozone signal instead. Only the BOPDMD and OptDMD-C methods are numerically stable over the entire 5-year integration, following and forecasting seasonal trends of the ozone response. These methods achieve forecast RMSEs of 1.88 and 1.11 tonnes (BOPDMD and OptDMD-C, respectively), while that of the other methods ranges from 1.26 (weak POD-SINDy) to 2.15 tonnes (POD-SINDy, Appendix Table B1). In all cases, the model discovery methods provide an improvement over forecasting for 1.5 years with the data-mean (RMSE of 2.32 tonne).



As altitude increases, the intricacy of the ozone response increases due to the higher complexity of local ozone chemistry and meteorological conditions, as well as the closer proximity to the perturbed emissions. Figure 4 shows that the seasonal behaviour seen near the surface is less influential at 16 km altitude, and almost non-existent at 26.0 km altitude. Lacking clear seasonal patterns, it also becomes more difficult for all methods to reproduce and forecast the ozone response in these domains. BOPDMD and OptDMD-C again produce the best reproductions of the original data, although they do not capture all features at these altitudes. These methods replicate the TAC204 data with RMSE of 17.28 and 10.41 tonnes for the 16 km grid cell, and 93.38 and 70.73 tonnes for the 26 km grid cell (BOPDMD and OptDMD-C respectively). When forecasting, these respective errors increase to 35.6 and 21.59 tonnes at 16 km, with 151.44 and 98.45 tonnes at 26 km.

These results highlight that accommodating all of these regimes within a single dynamical model is a significant challenge. For the near-surface grid cell, the ozone response is most stable and exhibits clear seasonal patterns, assisting its predictability. In this cell, forecasts from the data-driven methods are an improvement over propagating the mean in 71.4% of the tested cases, with the DMD-driven methods representing improvement in almost all cases. For the 16 and 26 km cells these fractions change to 69.4% and 59.2% respectively (Table B1) as forecasting future states becomes more difficult. At these altitudes, the DMD-driven methods still offer the most consistent improvements over the propagation of the temporal mean, but ultimately all methods have difficulties with the ozone response in the stratospheric grid cell. It may be that the data is just too complex at this altitude for these methods or that the size of the dataset may be insufficient to identify the appropriate large-scale features. Given the importance of this domain for this case study, this will negatively affect the ability to forecast in aggregated metrics such as the perturbation of ozone columns.

## 265 4.3 Zonal average



The differences in the ability to forecast and reconstruct different atmospheric domains are also apparent in the estimates of zonal average perturbations. The RMSE of the zonal average perturbation reconstructions and forecasts is presented in Table 1. We find that the DMD and POD-SINDy methods provide an improvement over the use of the temporal mean in most cases. When used for reconstruction the methods produce a lower RMSE than the propagation of the temporal mean in 81.6% of the tested combinations, and when forecasting this is the case for 75.5%. Of all methods tested, the OptDMD-C and BOPDMD variants produce the smallest overall RMSE in both reconstruction and forecasting the zonal average. Figure 5 shows that OptDMD-C consistently reduces the reconstruction RMSE by up to 60% compared to using the temporal mean, with reductions of -20% for forecasting. For BOPDMD these reductions are around -40% and -10% for reproducing and forecasting, respectively. Compared to the DMD methods, the POD-SINDy methods are less successful. They only provide notable (>10%) improvements over the use of the temporal mean in a few datasets, in some cases even yielding less accurate estimations. There are several factors that may affect the reduced performance of the POD-SINDy methods, which will be discussed in subsequent sections.


**Figure 5.** Change in the reconstruction (blue) and forecast (orange) RMSE with respect to reconstruction and forecasts using the temporal mean over all datasets for the methods used in this work. The left column shows the reconstruction of individual grid cells at altitudes of 1, 16, and 26 km (Table B1), the middle column the zonal average (Table 1), and the rightmost column the estimation of column ozone (Table 2). Markers with a black border denote the S1, S2, or S3 datasets.

The variability and the reconstruction and forecast errors of the methods applied to the TAC204 dataset are shown in Figure 6. The variability of the ozone response is localised around two areas of the stratosphere: the middle stratosphere and the area around the source of emissions. These areas coincide with the boundary between areas of net increases and depletion of stratospheric ozone, that are subject to seasonal variations. This variability pattern is generally captured by the DMD and POD-SINDy model, although their peak variability is lower because only low-ranking features are used in the reduced-order models. The effect of the lower variability is also visible in Figure 4, as the reconstructed signals do not exhibit the day-to-day variations of the original signal.

Figure 6 also shows that the methods have similar reconstruction errors despite their algorithmic differences. This suggests that the errors may predominantly be driven by sources outside of the models, such as the lack of higher-order features. The RMSE of the methods for the reconstruction and forecast of the atmosphere is presented in Table 1. Across all cases, we find

**Table 2.** RMSE (root mean squared error) of the reconstruction and forecast of mean changes in global column ozone for the DMD and POD-SINDy methods over the datasets in Dobson Units. The methods are fitted and tested for reconstruction (R) on 3.5 years of data, whereas the forecast (F) is tested on the subsequent 1.5 years of data. The data mean column shows the reconstruction and forecast errors if these were estimated using mean values from the fitting data. When method's RMSE is lower than the data mean approach, its values are printed in **bold**.

|            |   | DMD   | OptDMD | OptDMD-C | BOPDMD | POD-SINDy | POD-SINDy (ensemble) | POD-SINDy (weak form) | Data mean |
|------------|---|-------|--------|----------|--------|-----------|----------------------|-----------------------|-----------|
|            |   | 1     |        |          |        |           | (ensemble)           | (weak form)           | 1         |
| SAS204     | R | 0.08  | 0.08   | 0.06     | 0.06   | 0.12      | 0.12                 | 0.27                  | 0.09      |
|            | F | 0.28  | 0.28   | 0.29     | 0.29   | 0.23      | 0.21                 | 0.09                  | 0.28      |
| TAC204     | R | 0.10  | 0.12   | 0.06     | 0.09   | 0.17      | 0.17                 | 0.11                  | 0.17      |
|            | F | 0.27  | 0.29   | 0.23     | 0.3    | 0.24      | 0.25                 | 0.24                  | 0.30      |
| SAS162     | R | 0.04  | 0.04   | 0.02     | 0.03   | 0.05      | 0.05                 | 0.05                  | 0.05      |
|            | F | 70.07 | 0.07   | 0.06     | 0.07   | 0.06      | 0.06                 | 0.06                  | 0.07      |
| TAC162     | R | 0.05  | 0.04   | 0.02     | 0.04   | 0.06      | 0.05                 | 0.07                  | 0.04      |
|            | F | 0.09  | 0.09   | 0.10     | 0.08   | 0.05      | 0.04                 | 0.06                  | 0.09      |
| <b>S</b> 1 | R | 0.05  | 0.06   | 0.03     | 0.06   | 0.09      | 0.09                 | 0.09                  | 0.07      |
|            | F | 0.09  | 0.08   | 0.09     | 0.09   | 0.07      | 0.07                 | 0.05                  | 0.09      |
| S2         | R | 0.15  | 0.16   | 0.08     | 0.11   | 0.25      | 0.24                 | 2.14                  | 0.20      |
|            | F | 0.15  | 0.14   | 0.23     | 0.16   | 0.22      | 0.23                 | 1.44                  | 0.15      |
| <b>S</b> 3 | R | 0.03  | 0.03   | 0.01     | 0.02   | 0.04      | 0.04                 | 0.03                  | 0.03      |
|            | F | 0.05  | 0.05   | 0.03     | 0.07   | 0.06      | 0.06                 | 0.05                  | 0.05      |

that the model discovery methods improve the reconstruction over using the temporal mean in 40 out of 49 cases (81.6%). The OptDMD-C method always provides an improvement, whereas the percentage would be lower if only the POD-SINDy based methods are considered.

We also find that all methods exhibit similar error patterns when forecasting. In all forecasts, the methods underestimate the loss of the middle stratospheric ozone while overestimating the increases of the lower stratospheric ozone in the northern hemisphere. They also overestimate ozone depletion in the tropic upper-stratosphere. The consistency in these errors across the methods suggests that they may be caused by a common external factor, that cannot be forecast from the given data. This results in larger overall forecast errors, although in this application the data-driven methods are still better than propagating the temporal mean in 36 out of 49 cases (Table 1).


**Figure 6.** Standard deviation, reconstruction errors, and forecast errors of the zonal average ozone response. The top row shows the reference standard deviation, and mean values of the TAC204 dataset for the reconstruction and forecast period. The subsequent rows show the standard deviation and mean error for the reconstruction and forecast for the different methods. The standard deviation and errors are shown in tonnes of ozone.







Comparing the datasets with the local and global distribution of emissions, we do not find notable differences in the forecasting or reconstruction performance of the methods, despite the higher complexity of the latter datasets. Relative to reconstructing or forecasting with the temporal mean, there is no directly differentiable difference in method performance for the more complex datasets (Figure 5), although this may be because the accuracy of the temporal-mean estimate also reduces against more complex dynamics. Comparing the coefficient of determination, calculated by comparing all grid cells in the zonal average, we see that the coefficients are lower for the complex datasets (Figure C2). This shows that applying model discovery methods to these datasets is more challenging, but the difference is not big enough to affect the aggregated metrics.

#### 4.4 Derivative metric: Global ozone columns

In this section, we evaluate the ability to calculate derivative metrics, in this case the mean perturbation of the global ozone column in Dobson units (DU), from the zonal-average perturbations provided by the reduced-order models. This metric is an aggregate of the global ozone distribution, which is more sensitive to latitudinal misplacement of the ozone response because it depends on the surface area of the grid cells. The global mean ozone column response over time is shown in Figure 7 for the seven datasets, together with reconstructions and forecasts from the data-driven methods and their variants. The RMSEs of these reconstructions and forecasts are shown in Table 2.

We find that the ability of the data-driven methods to reconstruct and forecast the perturbation of the mean global ozone column varies considerably between the scenarios. Compared to propagating the temporal mean, the DMD and POD-SINDy methods provide better reconstructions in 55.1% of the applications. This is a reduction of 30.5% compared to the improvement of propagating the zonal average perturbation (Section 4.3). Figure 5 also shows that the RMSE reduction from the application of the methods also worsens upon calculation of the ozone columns. We expect that the reduction in accuracy is related to the higher sensitivity to latitudinal misplacement in the ozone column calculation. The POD-SINDy method and variants are most affected, producing higher RMSEs than the temporal mean in most cases. Only the OptDMD-C and BOPDMD methods still provide consistent improvements over the temporal mean for reconstruction, with OptDMD-C resulting in the largest reductions in error. Compared to the original data, both these methods have relative errors smaller than 10% in their reconstructions of the ozone column effect across all datasets.

Being an extension of the zonal averages, the ozone column estimates are also less accurate when forecasting future responses. These forecasts are better than propagating the mean in 55.1% of the applications. We also find that methods that reconstruct the data well do not necessarily provide the best forecasts (Figure 5). This suggests that, in some cases, such as the POD-SINDy method applied to the SAS204 data, the low forecast error may be attributable to coincidence more than the method. Despite this, the OptDMD-C and BOPDMD methods do forecast future trends when applied to some datasets, but the forecast values can deviate from the GEOS-Chem values. We expect that, like the drifts in zonal-average predictions, these shifts are also related to shifts in background conditions that are not foreseen by the methods.

Figure 7. Reconstruction and forecast of the change in global column ozone in DU over time for all datasets and methods. The vertical black line denotes the change from the reconstruction to forecasting domains, solid lines are reconstructions and dashed lines mark forecasts of future behaviour.

#### 5 Discussion

In this work, we present the evaluation of 7 variations of model discovery methods applied to 7 different datasets. However, the results that we present in the previous section do not represent the best results that we obtained. We have performed extensive








hyperparameter searches in which more than 50,000 configurations of the methods were fitted to different parts of the data (Appendix A). Through this approach, we found configurations with considerably smaller reconstruction and forecast errors for the ozone response than those we show in Section 4 (e.g., see Figure A3). We do not present these as our main results, as cherry-picking the best results does not necessarily paint an accurate picture of the performance that could be expected from these methods. Most prospective users of these methods may not have access to the future system behaviour to test the quality of forecasting, and they may also not have the resources to test tens of thousands of configurations. The results that we share in this work, therefore, are representative of the median performance of the hyperparameter search. For example, from 71 different configurations of the OptDMD-C method with 3.5 years of fitting data, we find a median forecast error of 16.7% for the S2 scenario, with a standard deviation 13.0%, and a range 6% to 75.4%. The OptDMD-C configuration presented in subsection 4.4 has a relative error of 16.4%. We present the median as we expect this level of performance is closest to what can be expected when these methods are applied to an untested dataset.

Overall, we find that the OptDMD-C or BOPDMD methods have the most robust performance in terms of forecasting and reconstruction of the data. This also holds true for the hyperparameter searches, where these methods generally perform best, although there are instances where DMD or unconstrained OptDMD perform better. The hyperparameter search has shown that there is no clear optimum for the rank of the DMD methods, with the exception of BOPDMD which becomes unstable at higher ranks (see Appendix A). Therefore, we do not expect that our results change significantly when the rank of the DMD methods is varied. For the application of DMD methods to other atmospheric chemistry datasets, we recommend the constrained OptDMD or BOPDMD methods first, because they most consistently yield low reconstruction errors while maintaining dynamical stability. The other methods we assess are able to yield similar results, but they are less consistent in numerical stability.

Compared to the DMD methods, we obtained worse performing reconstructions and forecasts of the reference data from the POD-SINDy based methods across all datasets. This has continually also been the case for the extensive searches that were run, and it is predominantly related to the difficulty of finding stable SINDy models for the time period that we assess. We expect this could be improved by the application of different SINDy extensions that are designed to promote model stability, such as the "Trapping SINDy" method by Kaptanoglu et al. (2021a), as our results show these stability-promoting approaches are very effective for DMD methods. In addition to the use of these extensions, it may also be possible that the SINDy methods may improve from further exploration of library model terms, as also discussed by Yang et al. (2024). We use a library with linear and quadratic terms, but this may not be well suited to this application. The use of different function libraries may improve results of this approach, although more work is necessary to identify what terms are most suitable. It could also be considered to integrate other data in the SINDy method than the POD time coefficients, such as the temperature or the distribution of other chemical species, as these might also directly affect the temporal coefficients.

Regardless of the method, we find that it is difficult to forecast future behaviour of the ozone response. We expect that this is not necessarily related to the methods, but a more fundamental difficulty with forecasting future behaviour of ozone. Ozone is





influenced by meteorological conditions, and shifts in these conditions may have considerable effects on its distribution. In our approach we attempt to predict ozone's behaviour from existing dynamics without integrating any meteorological information, and we show that this is a considerable challenge. We expect that better forecasts could be obtained if meteorological variables, such as global irradiance or temperature fields, are incorporated into the data provided to the methods. Alternatively, stochastic approaches, such as those proposed by Rao et al. (2024) and Boninsegna et al. (2018b), may also allow for better consideration of uncertainties due to changes in future conditions.

While our results show that all methods have some degree of error relative to the GEOS-Chem data, these methods all require 380 but a fraction of the computational cost and storage space footprint. The datasets that we use all require several gigabytes of storage each, and were created using hundreds of thousands of CPU-hours. Using the OptDMD-C method as an example, it can reproduce an estimate of the GEOS-Chem data in a matter of seconds while only needing 20 MB of storage space for the fitted model. This represents a reduction of several orders of magnitude in computational time and storage requirements. Furthermore, we have also shown that these reproductions can be accurate enough to calculate derivative metrics such as the change in global column ozone. However, it is likely that better ozone column models can be found by applying the methods to ozone column data directly, but the ability to do this conversion highlights the versatility of these methods.

#### 6 Conclusions and recommendations

Chemistry transport models (CTM) are critical in the evaluation of the environmental effects of emissions on our atmosphere, but their high computational cost and difficulty to use makes them impractical for implementation in multidisciplinary optimisation or operational and regulatory decision-making processes. Model discovery methods such as dynamic mode decomposition (DMD) and proper orthogonal decomposition (POD) paired with the sparse identification of non-linear dynamics (SINDy) can help to develop easy-to-use reduced-order models from chemistry transport model data. In this work, we evaluate the application of these methods on 7 datasets from case studies that assess changes in the global ozone distribution in response to supersonic aviation emissions over up to 10 years. Compared to earlier works, this represents a considerable increase in problem scope and complexity.

We show that these model discovery methods can be used to reconstruct and forecast datasets spanning entire atmospheres, with the constrained optimised DMD (OptDMD-C) and bagging optimised DMD (BOPDMD) methods being most effective. These methods can reconstruct and forecast the ozone response with small errors for several years without losing stability, resulting in substantial reductions in reconstruction (>50%) errors compared to estimations using the temporal mean. The reconstructed atmospheres of these methods can be accurate enough to calculate changes in the ozone column with relative errors smaller than 10% compared to the real data. Approaches based on DMD, unconstrained OptDMD, and POD-SINDy can also return reduced-order models with good performance, but these methods are less reliable because they do not promote stable solutions, affecting their dynamical stability over the multi-year timeframes. Ultimately, we find that suitable reduced-

- order models can be found with all methods, although the difficulty of finding stable configurations differs greatly between methods. Once stable models are found, we show that these are capable of reproducing CTM data with relative errors smaller than 5%, while reducing the computational and data storage requirements by several orders of magnitude and without requiring the same level of technical expertise as the CTM models. This capability may be valuable for several different applications, such as supporting decision-making and regulatory processes.
- 410 Code and data availability. The data and code supporting this work are publicly available on a 4TU.Researchdata repository, to be minted upon acceptance of this manuscript. Reviewers can preview this repository using the following link: https://data.4tu.nl/private\_datasets/QLtkjSKETpeO643uJMgNGUHd-qDQ0uQZicJ\_BfksYW0. The original datasets for the SAS204, SAS162, TAC204, and TAC162 scenarios are available through DOI:10.4121/d5947a0d-f34d-400b-87de-46ebda16ec44 (van 't Hoff et al., 2024a), and for the S1, S2, and S3 scenarios through DOI:10.4121/dd38833d-6c5d-47d8-bb10-7535ce1eecf1 (van 't Hoff et al., 2025a).
- 415 *Author contributions*. Conceptualization: ID, UF; Data curation: JH; Investigation: All; Methodology: All; Software: TC, UF, JH; Writing–original draft: TC, JH; Writing–review and editing: all
- Acknowledgements. This work was in part supported by the MORE and LESS (MDO and REgulations for Low-boom Environmentally Sustainable Supersonic aviation) research project as a part of Horizon 2020. Grant agreement 101006856, https://doi.org/10.3030/101006856. The GEOS-Chem simulations used for this work were supported by the Dutch national e-infrastructure with the support of the SURF Cooperative (Grant no. EINF-5945).

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

#### **Appendix A: Hyperparameter sweeps**

The usability of methods like POD and DMD can vary considerably depending on the hyperparameters that are used (e.g. rank, constraints). There is no clear best-practice with regards to these parameters, which is why as part of this work we ran extensive hyperparameter sweeps where thousands of variations of the methods were applied to different parts of the dataset. In this appendix, we shortly summarize the methodology and main results.

With this sweep, we primarily explored the effect of model ranks, different DMD variants, and the selection of fitting and forecast datasets. We find that there is no clear best practice with regard to the rank and variant of the DMD method, but some trends can be identified (Figure A1). For the conventional DMD method and unconstrained OptDMD, the median performance and the variance between the methods is not affected by the rank. For the constrained OptDMD and BOPDMD the median forecast accuracy and its spread increase with higher ranks. The latter in particular exhibiting increasing errors above the rank of 8.

Concerning the size of the dataset, we find that DMD and OptDMD generally work better with small datasets (<500 datapoints) whereas BOPDMD methods require at least several years of data (>1000 datapoints) to achieve similar performance (Figure A2). Across all methods the median forecast error and its spread decrease when the methods have access to more data.

Regardless of the rank of the method and the length of the fitting data, DMD models can be found that have forecast errors smaller than 10%, but the odds of finding such models improve under the conditions mentioned above. Regardless of the settings, no methods have better relative forecast errors than 5%, which may represent a limit for what can be expected of DMD methods for data as variable as daily changes in ozone concentrations.

In the end, we do not find clear suggestions for the best-practice application of DMD methods to ozone perturbation data.

Our results suggest that the most suitable DMD method and its configuration depend primarily on the data on which it is applied, and therefore we recommend that prospective users of these methods explore different configurations as well for their own data.

Figure A1. Results of hyperparameter sweep of DMD methods over the DMD rank, for the TAC204, TAC162, S1, and S2 emission scenario datasets. The solid lines indicate the median  $O_3$  column change forecast error in percent. The shaded areas mark the full range of the error distribution, and the dotted lines mark the lowest forecast error achieved by any of the tried configurations.

Figure A2. Results of hyperparameter sweep of DMD methods over the DMD rank, for the TAC204, TAC162, S1, and S2 emission scenario datasets. The solid lines indicate the median  $O_3$  column change forecast error in percent. The shaded areas mark the full range of results, and the dotted lines mark the lowest forecast errors achieved by any of the tried configurations.

**Figure A3.** Selection of DMD models identified by the hyperparameter sweep, selected for the best 2-year forecast performance for the SAS204, TAC204, TAC162, and S1 scenarios. The rank and the length of fitting data is allowed to vary between the configurations, therefore the selected best results have different ranks and ranges of fitting data. In some cases however, such as for OptDMD with the TAC162 scenario, suitable configurations could still not be found.

# **Appendix B: Supplementary Tables**

**Table B1.** RMSE (root mean squared error) of the reconstruction and forecast of mean changes near-surface ozone mass in tonnes for the DMD and POD-SINDy methods over the datasets. The Data mean column shows the reconstruction and forecast errors if these were estimated using mean values from the fitting data. Values printed in **bold** are lower than the data-mean estimate.

| SAS204 R 1.13 1.55 F 1.75 2.55 TAC204 F 1.73 2.15 SAS162 F 1.28 1.65 TAC162 R 1.05 1.34 TAC162 F 1.59 1.5 S1 R 0.16 0.16 S2 R 0.68 0.66 S2 R 0.69 0.67 | 3 1.14<br>7 0.57<br>3 1.11<br>3 0.92<br>5 1.1<br>4 0.54<br>0 0.87 | 1.33<br>2.46<br>0.82<br>1.88<br>1.07<br>1.55<br>0.7 | 1.88<br>3.74<br>1.59<br>2.15<br>1.47<br>1.73 | 2.25<br>2.42<br>1.58<br>2.16<br>1.46 | 3.2<br>2.56<br>0.97<br>1.26<br>1.65 | 2.13<br>2.68<br>1.9<br>2.32<br>1.75 |
|--------------------------------------------------------------------------------------------------------------------------------------------------------|-------------------------------------------------------------------|-----------------------------------------------------|----------------------------------------------|--------------------------------------|-------------------------------------|-------------------------------------|
| TAC204 R 0.95 1.22  F 1.73 2.13  SAS162 R 1.12 1.33  TAC162 F 1.28 1.66  TAC162 F 1.59 1.9  S1 R 0.16 0.16  F 0.28 0.27  R 0.68 0.66                   | 7 0.57<br>3 1.11<br>3 0.92<br>5 1.1<br>4 0.54<br>9 0.87           | 0.82<br>1.88<br>1.07<br>1.55<br>0.7                 | 1.59<br>2.15<br>1.47<br>1.73                 | 1.58<br>2.16<br>1.46                 | 0.97<br>1.26<br>1.65                | 1.9<br>2.32                         |
| TAC204 F 1.73 2.13  SAS162 R 1.12 1.33  F 1.28 1.65  TAC162 F 1.59 1.3  S1 R 0.16 0.16  S1 F 0.28 0.2  R 0.68 0.66                                     | 3 1.11<br>3 0.92<br>5 1.1<br>4 0.54<br>9 0.87                     | 1.88<br>1.07<br>1.55<br>0.7                         | 2.15<br>1.47<br>1.73                         | 2.16<br>1.46                         | 1.26<br>1.65                        | 2.32                                |
| SAS162 R 1.12 1.33 F 1.28 1.65 TAC162 R 1.05 1.34 F 1.59 1.9 S1 R 0.16 0.16 F 0.28 0.27 R 0.68 0.66                                                    | 3 0.92<br>5 1.1<br>4 0.54<br>9 0.87                               | 1.07<br>1.55<br>0.7                                 | 1.47<br>1.73                                 | 1.46                                 | 1.65                                |                                     |
| SAS162 F 1.28 1.66  TAC162 R 1.05 1.34 F 1.59 1.59 S1 R 0.16 0.16 F 0.28 0.27 R 0.68 0.66                                                              | 5 1.1<br>4 0.54<br>9 0.87                                         | 1.55<br>0.7                                         | 1.73                                         |                                      |                                     | 1.75                                |
| TAC162 R 1.05 1.36  F 1.59 1.9  R 0.16 0.16  S1 F 0.28 0.22  R 0.68 0.66                                                                               | 0.54<br>0 0.87                                                    | 0.7                                                 |                                              | 1.76                                 |                                     |                                     |
| TAC162 F 1.59 1.5<br>R 0.16 0.16<br>F 0.28 0.22<br>R 0.68 0.62                                                                                         | 0.87                                                              |                                                     | 1 33                                         |                                      | 1.55                                | 1.75                                |
| S1 R 0.16 0.14<br>F 0.28 0.22<br>R 0.68 0.62                                                                                                           |                                                                   | 1.25                                                | 1.55                                         | 1.17                                 | 1.46                                | 1.91                                |
| F 0.28 0.27<br>R 0.68 0.62                                                                                                                             | 0 14                                                              | 1.20                                                | 2.52                                         | 2.0                                  | 2.0                                 | 2.01                                |
| R 0.28 0.2<br>R 0.68 0.62                                                                                                                              | , ,,,,,,                                                          | 0.18                                                | 0.18                                         | 0.18                                 | 0.19                                | 0.17                                |
| \$2                                                                                                                                                    | 0.36                                                              | 0.28                                                | 0.22                                         | 0.22                                 | 0.22                                | 0.26                                |
| \$2                                                                                                                                                    | 0.29                                                              | 0.38                                                | 0.87                                         | 0.89                                 | 29.69                               | 0.75                                |
| F   0.89 <b>0.7</b> 3                                                                                                                                  | 0.91                                                              | 0.74                                                | 0.64                                         | 0.62                                 | 16.46                               | 0.74                                |
| R 0.15 0.16                                                                                                                                            | 0.11                                                              | 0.11                                                | 0.27                                         | 0.27                                 | 0.18                                | 0.19                                |
| S3 F 0.24 0.23                                                                                                                                         | 0.25                                                              | 0.34                                                | 0.23                                         | 0.23                                 | 0.23                                | 0.23                                |
| 16 km altitude                                                                                                                                         |                                                                   |                                                     |                                              |                                      |                                     |                                     |
| R   20.61 21.21                                                                                                                                        | 16.53                                                             | 18.88                                               | 27.95                                        | 31.13                                | 26.58                               | 24.55                               |
| SAS204 F 21.08 24.53                                                                                                                                   |                                                                   | 29.24                                               | 38.99                                        | 29.74                                | 24.02                               | 25.87                               |
| R 187 201                                                                                                                                              |                                                                   | 17.25                                               | 25.12                                        | 25.48                                | 18.32                               | 28.5                                |
| TAC204 F 24.96 27.62                                                                                                                                   |                                                                   | 28.44                                               | 28.4                                         | 28.54                                | 23.87                               | 29.99                               |
| R 49.9 54.60                                                                                                                                           |                                                                   | 35.6                                                | 68.19                                        | 67.44                                | 73.62                               | 74.29                               |
| SAS162 F 57.41 69.47                                                                                                                                   | 32.08                                                             | 75.21                                               | 75.1                                         | 76.57                                | 66.99                               | 71.6                                |
| R 21.23 20.69                                                                                                                                          |                                                                   | 17.68                                               | 20.77                                        | 24.64                                | 21.46                               | 21.51                               |
| TAC162 F 21.69 21.74                                                                                                                                   |                                                                   | 15.63                                               | 24.22                                        | 19.12                                | 19.82                               | 21.74                               |
| R 1.52 1.59                                                                                                                                            | 1.33                                                              | 1.88                                                | 2.2                                          | 2.2                                  | 2.06                                | 2.09                                |
| S1 F 2.69 2.63                                                                                                                                         | 3.5                                                               | 2.39                                                | 2.02                                         | 2.01                                 | 1.74                                | 2.59                                |
| R 6.87 6.47                                                                                                                                            | 3.2                                                               | 4.13                                                | 9.56                                         | 9.91                                 | 163.65                              | 7.77                                |
| S2 F 9.6 <b>8.5</b> 0                                                                                                                                  | 10.29                                                             | 8.52                                                | 7.53                                         | 7.13                                 | 90.96                               | 8.78                                |
| R 2.48 2.50                                                                                                                                            | 1.35                                                              | 1.54                                                | 3.16                                         | 3.16                                 | 3.18                                | 3.1                                 |
| S3 F 3.39 3.89                                                                                                                                         | 3.5                                                               | 5.46                                                | 3.8                                          | 3.8                                  | 3.86                                | 3.94                                |
| 26 km altitude                                                                                                                                         |                                                                   |                                                     |                                              |                                      |                                     |                                     |
| R   160.58 161.54                                                                                                                                      | 131.64                                                            | 115.96                                              | 150.16                                       | 172.15                               | 324.82                              | 168.98                              |
| SAS204 F 122.34 131.99                                                                                                                                 |                                                                   | 216.17                                              | 139.0                                        | 151.08                               | 226.53                              | 140.97                              |
| R 85 39 87 23                                                                                                                                          |                                                                   | 93.38                                               | 105.56                                       | 102.93                               | 105.63                              | 107.21                              |
| TAC204 F 96.97 95.49                                                                                                                                   |                                                                   | 151.44                                              | 86.99                                        | 86.88                                | 106.82                              | 96.4                                |
| R 26.94 29.6                                                                                                                                           |                                                                   | 21.03                                               | 30.7                                         | 30.53                                | 31.96                               | 31.26                               |
| SAS162 F 25.23 23.51                                                                                                                                   |                                                                   | 29.05                                               | 28.7                                         | 28.19                                | 23.17                               | 24.48                               |
| R 39.6 40.0°                                                                                                                                           |                                                                   | 29.28                                               | 40.14                                        | 39.26                                | 41.35                               | 41.59                               |
| TAC162 F 28.69 31.5                                                                                                                                    |                                                                   | 26.12                                               | 40.4                                         | 34.11                                | 32.8                                | 33.45                               |
| R 14.84 14.59                                                                                                                                          |                                                                   | 10.13                                               | 14.61                                        | 14.63                                | 15.32                               | 20.3                                |
| A1 F 12.01 18.58                                                                                                                                       |                                                                   | 16.11                                               | 15.19                                        | 14.63                                | 17.11                               | 19.84                               |
| R 70.3 64.7                                                                                                                                            |                                                                   | 41.5                                                | 84.08                                        | 87.37                                | 3915.06                             | 79.85                               |
| A2 F 84.21 <b>80.7</b> 8                                                                                                                               |                                                                   | 66.0                                                | 75.41                                        | 76.63                                | 2134.65                             | 83.19                               |
| R 7.07 6                                                                                                                                               |                                                                   | 3.34                                                | 8.98                                         | 8.98                                 | 8.74                                | 8.17                                |
| A3 F 10.27 9.11                                                                                                                                        |                                                                   | 3.36                                                | 11.89                                        | 11.89                                | 8.51                                | 9.44                                |

# **Appendix C: Supplementary figures**

**Figure C1.** Altitude of the center of mass of the ozone response in the TAC204 dataset over time, calculated through a weighted average of the grid cells. Top row shows the northern hemisphere  $(20\circ\text{N} \text{ to } 85^\circ\text{N})$ , the middle shows the tropics  $(-20\circ\text{N} \text{ to } 20^\circ\text{N})$ , and the bottom row shows the southern hemisphere  $(-85\circ\text{N} \text{ to } -20^\circ\text{N})$ .

**Figure C2.** Comparison of the coefficient of determination for the zonal-average ozone response reconstructions (blue) and forecasts (orange) across all datasets for the individual methods.