# Peer review of "Data-driven discovery and model reduction methods for the atmospheric effects of high altitude emissions"

_EGUsphere, 2025_

## Author Comment (AC2)

Dear Editor and referees,

Thank you for arranging this review of our work. We have carefully considered and addressed the reviewers' feedback and believe that this has considerably improved the quality and presentation of our work. In this document you will find our response to the reviewers' comments, answering their questions and specifying how their feedback is integrated into the revised manuscript. Reviewer comments are shown in *italics*, and our response is in **bold**. Page, line, or figure numbers refer to the updated manuscript without tracked changes.

The revised manuscript includes the addition of a new variation of the BOPDMD method suggested by reviewer #2, and textual additions to add clarifications to the manuscript. During the review process we also found out that development pySINDy library that our initial work relied on is no longer publicly available. To ensure our work is reproducable we have updated our code, and the associated repository, to recent public versions of the pySINDy (v2.0.0) and pyDMD (v2025.03.01) libraries. Due to underlying changes in the code of these libraries there are numerical changes to our results. Notably, the OptDMD and weak SINDy algorithms are more stable and perform better with this new version. Despite numerical changes throughout the revised manuscript, our main conclusions do not change significantly.

Thank you again for considering our submission to *Geoscientific Model Development*, we look forward to hearing from you in due course.

Kind regards,

Jurriaan van 't Hoff, Tom van Cranenburgh, Urban Fasel, and Irene Dedoussi.

**Reviewer RC1**

*### General comments ###*

*The paper analyses different methods to develop surrogate models for the effect of different supersonic aviation scenarios on the global ozone concentration based on heavy chemistry transport model computations. The core idea of the paper to use surrogate models for atmospheric modelling to enable an integration of environmental effects into processes, where extensive computations are infeasible, like iterative design optimizations, is very relevant. Testing DMD- and POD-SINDy-based approaches on existing datasets for the exemplary case of ozone concentration changes can support scientific progress in the field and help fellow researchers in the design of future studies and on the application of the examined approaches, even though the choice and suitability of DMD and POD-SINDy for the aimed application seems questionable.*

*In general, the paper gives a good overview over the used methods and describes and discusses the results well structured, ordered and clear. In some parts there is a focus on subparts of the study, leaving other parts of similar importance behind or lacking a summary of the remaining parts. In addition, some assumptions and basic settings chosen lack fully convincing explanations.*

**Thank you very much for your thorough and detailed review. We have used your comments and suggestions to make significant improvements to the manuscript. Please see our responses to your specific comments below for further information.**

*### Specific comments ###*

*In the choice of the study setup as a hyperparameter search picking the median performing model as the "result" the mentioned point becomes the clearest.*

*Doing a hyperparameter search is clearly necessary due to a lack in best-practice settings, as mentioned in Appendix A. Therefore, it is recommended on page 24, to do such a search for new, similar studies with another dataset. Now the question comes up, why this study uses only the median model of the hyperparameter search as a representative result, if new studies should also do a hyperparameter search and could apparently choose their hyperparameters optimal according to the search. Therefore, the models with optimal hyperparameters at least for the reconstruction data should be taken as reference or be discussed as well.*

*As an explanation for the chosen median model, the paper says on page 18, that prospective users will not have access to future data to assess the forecasting performance and thereby they will not be able to find the optimal forecast model. It is correct, that there will be no reference data to validate against in many cases, but still to my understanding the forecasting quality can be assed just as done in this study, by excluding e.g. the last one and*

*a half years of the existing data (parallel to test/train split in ML). Then the settings of the model with the best forecasting performance could be used to train a new model based on the full dataset and I would expect a comparable forecasting quality for the first one and a half future years. And even if no split into a train/test dataset is possible dues to a lack of data, still a hyperparameter search for the reconstruction data could lead to a far better model, than just the median. I would find it interesting to have a look at such options. Never the less, the comparison of the different approaches based on the median model still gives a good overview over their performance and characteristics, even though not showing the full potential.*

**Thank you for this feedback and the provided suggestions. We performed some tests to explore the full potential of the of the models, as suggested. Notably, we found that the selection of models based on the best reproduction resulted in less accurate forecasts in most cases, because this selection criterion promotes higher-rank methods that overfit to the data. The ideal forecasting model may therefore need to balance reconstruction and forecast capabilities with the availability of data, which requires more thorough exploration to make definitive conclusions. However, we realize that this information is of interest to the readers, and we have made amendments to the manuscript to reflect this and to broach the topic of model selection strategies for future work (Lines 363-367).**

*As the goal of this study you mention the usability in applications like MDO. In my understanding the models you created and the methods you use show very limited usability in this field, as for MDO the surrogate model needs to be able to do predictions based on changed inputs (e.g. changed propulsion emissions, changed AC geometry, changed routes). The models you created are able to reconstruct data from chemistry transport models and do extrapolations/forecasts based in the data, but do not feature the possibility to input the mentioned changes into the model. Therefore, it seems questionable, if the goal of this study can be reached with the chosen methods.*

**This is a fair comment as the capabilities of these methods as they are applied in this work indeed do not directly allow for the evaluation of factors such as new aircraft geometries. Our work represents a step towards that penultimate goal. We consider that it is first necessary to establish to what extend these methods may model atmospheric responses to emissions in coupled tropo- and stratospheric systems, since these methods have not been previously applied in the context of evaluating changes in the atmospheric composition triggered by changes in the emissions, and that is the goal of this manuscript. We have made some additions to the introduction (lines 59-63) and discussion (lines 413-417) to clarify this vision.**

*Section 4.2 focuses in the TAC204 dataset. It does not become clear why particularly TAC204 is chosen, as the global scenarios are assumed to be more complex and thereby the more critical scenario to look at. The findings from the other datasets are not mentioned and*

*not summarized, leaving the question, if their results are mostly the same as for TAC204 or not.*

**The TAC204 dataset was chosen because it is the dataset with the largest ozone response out of those where there are no emissions in the grid cells of interest (TAC204, TAC162). The atmospheric response in the selected grid cells (1.1 km, 18.3 km, and 26 km, representing troposphere, tropopause, and stratosphere, respectively) is therefore driven by transport and meteorology and less so by directly emitted emissions. We realize that this is not explained in the text and have amended it accordingly (Lines 242-244). Furthermore, we have also added a summary of the grid cell modelling in the other datasets (Lines 280-281, Figures C3-C8).**

*The first part of the results section (4.1) focuses on the interpretation of the surrogate models with their different spatial modes. Here a more detailed qualitative look into the single atmospheric dynamics covered and also those not covered would be great in my eyes, even though it is mentioned, that through the combination of effects a direct attribution is challenging. But e.g. understanding which dynamics are not covered, could help in the assessment of the capabilities of DMD and POD-SINDy for future studies (just a few examples). A summary of these interpretations could also be included in the discussion section.*

**Thank you for the suggestion. While we cannot separate with confidence cause and effect, we have added some explanation of our hypotheses on the involved dynamics (lines 226-228, 230-231).**

*The goal of the study is according to the introduction the creation and examination of models, that can be used for MDO, etc. In the conclusion the step back to the level of looking at the application of the models and the implications arising from that is missing. Are models of the quality you found usable for MDO or is the quality most likely insufficient or only usable for the main huge effects?*

**Thank you for making us aware of this gap in the conclusion. We have expanded it accordingly (Lines 440-445). We consider that, at the current state, some of the methods are sufficiently accurate for efficiently reconstructing temporal trends to allow them to be embedded in processes such as MDOs to allow for more detail in modelling over the temporal mean that may otherwise be used. However, more value could be extracted from these models by improving their forecasting accuracy or by developing methods to combine multiple dynamic response models (e.g., at different cruise altitudes) to predict responses in unassessed cases (e.g., intermediate cruise altitudes)**

*In the conclusion section the quality of the models is mostly evaluated based on reconstruction results, but not the forecast, which is at least equally relevant for many applications. Please include forecast results here as well.*

**We have amended the conclusion to also summarize the forecast results (Lines 439-440)**

*There are two points I see as interesting for future studies and I am interested in what you think about them or whether you are planning to continue your research into that direction.*

*First is, if you think some kind if informed DMD could lead to reasonable improvements, because one could feed physics and additional information into the models. The second one you already mentioned in your discussion (page 18). You wrote that you do only use linear and quadratic terms for POD-SINDy and better results might be achieved with further terms. I want to emphasize this point, as the complex dynamics of the atmosphere might not be easily depicted by first and second order polynomials. I would be very interested in seeing the results considering more complex terms.*

**Thank you for these suggestions – these both could be very interesting for follow-up research. Our results already show that the performance of DMD models can improve greatly with more specialized variations, and we similarly believe much could be gained by continuing to develop on this. Additionally, the ozone response strongly depends on a range of environmental conditions (temperature, radiative flux, transport patterns) that are not included in the data provided to the methods, and an interesting direction would be to consider embedding these variable fields (e.g. meteorological data) in the data provided to the methods. It may indeed also be possible to develop on these methods by embedding constraints inspired by physics or atmospheric transport processes within the algorithms themselves.**

**As also mentioned in the manuscript, we agree that we likely do not achieve the full potential of POD-SINDy based methods as we only use relatively simple terms. In the work leading up to this manuscript we briefly experimented with more complex terms, such as sinusoidal functions, that may fit better with the seasonal variability of the data. Exploring these alternative library terms could also constitute interesting next steps.**

*### Technical corrections ###*

*- p.1, l.12: "... DMD method reduces ...": Reduces compared to what?*

**This is compared to the propagation of the spatio-temporal mean. It has been clarified (Line 13).**

*- p.2, l.27: "... to integrate their environmental ...": The reference of "their" is not fully clear.*

**"Their" referred to the engineers used as example in this sentence and has now been removed (Line 28).**

*- p.3, l.64: "… 8 Tg …": 8 Tg of what - CO2/NOx/Total?*

**The line refers to the mass of annual fuel burn of supersonic aircraft. We have added a line to explain this better (Line 70)**

*- p.5, Figure 2: Unit of z-axis missing.*

**We have added the missing unit to the caption.**

*- p.6, l. 122: "… spatial and temporal coefficients …": In the sentence before you wrote spatial mode and temporal coefficient. Please be precise and consistent in the naming.*

**Corrected to match the previous sentence (Line 127)**

*- p.6, l.125: "… to be forecast, …": I think it should be "forecasted" here.*

**Corrected.**

*- p.6, l.144: "… sequential grid search …": I did not find the parameters for the grid search in the paper. It might be helpful to see which parameters and which options you used.*

**The parameters for the grid search have been added. Lines 151-152**

*- p.7, l.158: The acronym DMD is introduced a second time.*

**Second introduction has been removed.**

*- p.9, Figure 3 – caption: "… OptDMD-C methods (left) …": Should be "method" only; "… shown as as pairs …": Delete one as.*

**Corrected.**

*- p.14, Table 2 - caption: Introduce acronyms R & F.*

**We moved the introduction of the acronyms to the first line, matching the caption of Table 1.**

*- p.16, l.318 & l.328: In both lines, one for reconstruction, one for forecast, 55.1% are better than mean. Is this maybe a copy and paste error?*

**Thank you for marking this, this is no longer the case in the revised document.**

*- p.19, l.399, l.403: "… small errors …", "… good performance …": What does small/good mean here. Please concretize.*

**We have changed these lines to use concrete values (lines 430-433)**

*- p.19, l.404: "… suitable …": Please concretize, suitable for what and why.*

**We have replaced this with a more concrete explanation, suitable was not the right choice of words (Lines 436-437).**

*- p.23, l.509: "… no clear best practice with regards to these parameters, …": Please include reference/source.*

**This statement was not derived from any particular source, it was rather a reflection on the limited availability of reference works similar to this to derive best-practice hyperparameters from. We have changed the lines in question to include relevant literature and be less of a blanket statement (lines 556-557).**

*- p.25 Figure A2 – caption: "… over the DMD rank, …": Copy and paste error, should be "over the fitting dataset length".*

**Corrected.**

*- Figure A1, A2, A3: Why do you not include the results from all datasets?*

**In these figures we only included a selection of datasets because all datasets show similar trends and have overlapping areas if plotted in this manner. Therefore, we found showing all datasets hinders the visibility of the figure.**

*- p.27, Table B1 - caption: Introduce acronyms R & F.*

**Introduction has been added.**

*- p.29, Figure C2: Marker for reconstruction is circle in figure, but cross in legend.*

**The legend has been corrected.**

**Reviewer RC2**

*1) Scientific significance*

*The manuscript presents an application of reduced-order modelling (DMD variants and POD-SINDy) to complex chemistry transport models (GEOS-Chem).*

- *However, the introduction should reference Linear Inverse Modeling (LIM) to better contextualize the approach within existing DMD-based atmospheric science methods.*

**Thank you for your detailed review. The original manuscript indeed overlooks linear inverse modelling and this is important for contextualizing our work. We have amended the introduction accordingly (Lines 43-45)**

- *The discussion must explicitly address the physical implications of modeling this system via DMD/POD-SINDy, perhaps regarding the stationarity of the time series and whether properties like physical conservation laws (e.g., ozone mass conservation) are obeyed.*

**We have amended the discussion of the manuscript to include the physical implications of modeling using these methods (Lines 396-398)**

*2) Scientific quality*

*The approach is valid, but specific methodological choices lack justification.*

- *The authors should explain why only "low-order polynomial terms" were selected for the SINDy library (line 365)*

**Low-order polynomials were selected for this work because they are straightforward to implement, recommended as starting point for new applications (Fasel et al. 2022), and because they have shown to be successful in other work (Yang et al. 2024). More extensive exploration of terms has been briefly considered (as also discussed in response to RC1) and would be interesting to pursue in future work. We have added an explanation to this paragraph (Line 389-390)**

- *Additionally, were different time lags and time delay embeddings tested?*

**Time lags were tested in preparation of this work, and we found that it did not yield considerable benefit. Embeddings were not tested, and we consider that further exploration may be worthwhile. We have added this information to the main text (Lines 380-381)**

- *The hyperparameter search involving "50,000 variations" appears excessive; a heuristic approach or clearer explanation of the grid search density is needed.*

**We understand that the source of this number is not clear without additional explanation. We have added an explanation of the grid search to the appendix (lines 561-563). The grid search has also been redefined in the revision to integrate the new BOPDMD variant while we updated our code to publicly available library versions. With the new definition we reduce the density of grid points in dimensions where the first hyperparameter search showed high-resolution exploration was unnecessary.**

- *Additionally, could the authors kindly justify why BOPDMD requires more data than the other DMD methods (line 520).*

**The BOPDMD trains an ensemble of DMD models on a random sub selection of 20% of the dataset via bagging. With datasets smaller than 1000 days this leaves less than 8 months of daily samples for the models in the ensemble, and this might not be sufficient for them to properly capture the annual and multiannual trends in the data. We have added an explanation of this in the text (lines 571-574). We expect that this may be less prevalent when the base dataset is larger, or the trial size for BOPDMD less small.**

- *Please justify how using the derivative for modeling SINDy make it "susceptible to noise"(ine 150).*

**In the data that we use in this work the 'noise' is primarily in the form of the day-to-day variations in mean daily ozone response, which occur as instantaneous changes due to the daily temporal resolution. These are of sufficiently large magnitude in relation to the longer-term ozone trends to also show up as a noisy signal in the derivative, which makes it more difficult for the SINDy algorithm to fit a dynamical model as it will fit to account for these variations. This is however a problem with the data that we use, and not the SINDy method necessarily. We have changed this line of text accordingly (Lines 154-156)**

- *Is it possible to constrain the eigenvalues in BOPDMD to offer yet another DMD variant?*

**The results that were shown for the BOPDMD algorithm were already constrained to stable eigenvalues; however we understand that this was not well communicated in the manuscript. We have changed the BOPDMD naming to C-BOPDMD to match the new naming of C-OptDMD and added an unconstrained BOPDMD variant for comparison. (Changes throughout text, additions to tables and figures)**

*3) Scientific reproducibility*

*Would the authors kindly release the code used for the experiments in this paper? This is an essential step to ensuring scientific reproducibility. Some smaller comments include:*

**The code necessary to reproduce the results in this work are already available in the preview of the data repository linked in the Code and Data availability. In-line with the chief editor's comment, this has now been made public and is accessible through DOI: https://doi.org/10.4121/d7c8091b-fc2b-4c21-a498-d4a01c9a7a40. It is included in the revised manuscript (lines 447-449)**

- *Why are 22 degrees North selected in (line 225).*

This area in general was selected because this coincides with the latitude of emissions of the SAS204 inventory, and because it is a latitude with challenging transport dynamics.

- *What are the drawbacks of flattening latitude and altitude dimensions?*

Within the timescales used for studying the environmental effects of supersonic aviation, the data is often flattened to latitude and altitude because longitudinal mixing is much faster than in the other two dimensions, producing data that is nearly homogeneous in the longitudinal direction when assessed over yearly timescales. The methods we use could however also be applied to unflattened data as well. A drawback of this flattening is that some information (e.g. longitudinal mixing trends) is lost in the preprocessing, but we do not expect this to have large effects on the case study we cover in this manuscript.

- *Would the authors consider verifying the numerical stability of the DMD models using their eigenvalues?*

We have added a figure to the appendix which shows that the eigenvalues of all models used in the revised manuscript (Figure C2). For all models the eigenvalues fall on the unit circle. We have also added a summary to the main body of text to augment the discussion of model stability (Line 253-258).

*4) Presentation quality*

*The work is well presented and organized. There are only some small comments.*

- *Notation requires correction; Equation 1 should use boldface for vectors and arguably represent a model rather than just data structure.*

Corrected. We have made a small correction to line 111 to suggest this is a model rather than a data structure.

- *The naming convention "OptDMD-C" is confusing (implies control rather than constraints). Perhaps use "C-OptDMD?"*

We have changed the text to use C-OptDMD as suggested by the reviewer (Changes throughout text and figures).

- *Would the authors kindly be consistent with using (-) in (e.g., 60% vs -40% and -10%). (line 270)*

We have reviewed all instances and updated them to be consistent.

**References:**

Fasel, U., Kutz, J. N., Brunton, B. W., and Brunton, S. L.: Ensemble-SINDy: Robust sparse model discovery in the low-data, high-noise limit, with active learning and control, Proceedings of the Royal Society A: Mathematical, Physical and Engineering Sciences, 478, https://doi.org/10.1098/RSPA.2021.0904, 2022.

Yang, X., Guo, L., Zheng, Z., Riemer, N., and Tessum, C. W.: Atmospheric Chemistry Surrogate Modeling With Sparse Identification of Nonlinear Dynamics, Journal of Geophysical Research: Machine Learning and Computation, 1, e2024JH000 132, https://doi.org/10.1029/2024JH000132, 2024.